# Comparative Study by Systematic Review and Meta-Analysis of the Peri-Implant Effect of Two Types of Platforms: Platform-Switching versus Conventional Platforms

**DOI:** 10.3390/jcm11061743

**Published:** 2022-03-21

**Authors:** Alejo Juan-Montesinos, Rubén Agustín-Panadero, Maria Fernanda Solá-Ruiz, Rocío Marco-Pitarch, Jose María Montiel-Company, Carla Fons-Badal

**Affiliations:** 1Independent Researcher, 46010 Valencia, Spain; alejo.juan.montesinos@gmail.com; 2Department of Oral Medicine, Faculty of Medicine and Dentistry, University of Valencia, 46010 Valencia, Spain; m.fernanda.sola@uv.es (M.F.S.-R.); rociomarco1@hotmail.com (R.M.-P.); jose.maria.montiel@uv.es (J.M.M.-C.); carlafonsbadal@gmail.com (C.F.-B.)

**Keywords:** prosthetic platform change, dental implants, meta-analysis, randomized clinical trials, vertical bone loss

## Abstract

Objective: The aim of the systematic review and meta-analysis carried out was to evaluate the effects of changing the prosthetic platform on peri-implant tissue after 1 year of prosthetic loading. Material and methods: In November 2020, an electronic search was carried out in PubMed, EMBASE, Web of Science, and Scopus databases with the aim of obtaining all the randomized clinical trials that had been published in the preceding 10 years comparing the effects on the peri-implant tissue of implants with a prosthetic platform change and implants with a conventional platform for at least 1 year after prosthetic loading. Randomized model meta-analyses of the selected studies were performed to compare the results of the two implant groups in terms of vertical maintenance of bone level and increased probing depth. Results: Nine studies were included, summing up a total of 475 implants with prosthetic platform exchange and 462 implants with a conventional platform. Implants with prosthetic platform exchange had less peri-implant bone loss than implants with a conventional platform (mean difference of 0.255 mm, statistically significant) but suffered a greater increase in probing depth (mean difference of 0.082 mm, not statistically significant). However, the probing depth from One Study Remove revealed a statistically significant increase of 0.190 mm in the prosthetic platform exchange group compared to the conventional platform group. Conclusion: Implants with platform switching suffer less peri-implant bone loss after 1 year of loading than implants with a conventional platform. Further long-term studies are required to observe how these differences vary over time.

## 1. Introduction

The maintenance and stability of the marginal bone level of dental implants have been considered a critical factor for the preservation of peri-implant soft tissues, as well as to avoid further bone loss around the implant and its possible progression toward peri-implantitis [1].

The daily technical advances in the field of implant dentistry have significantly reduced physiological bone remodeling surrounding dental implants, which occurs mainly during the first year after implant placement and is influenced by various biological, technical, and surgical factors [2].

Classically, the most commonly used implant type was the two-part implant with an external hexagon connection [3,4]. Several studies have identified a physiological bone loss surrounding this type of implant so frequently that Albrektsson defined a vertical physiological bone loss of 1.5 mm after the first year and 0.2 mm for each subsequent year as a success criterion for these implants [5].

Later, Lazzara and Porter used larger-diameter implants with conventional diameter abutments and observed less peri-implant bone loss using combination, which resulted in the development of the prosthetic platform-switching technique [6]. The different hypotheses proposed to explain the lower peri-implant bone loss in implants with prosthetic platform switching were the following: the horizontal separation distance it creates between the microgap and the peri-implant bone, the possibility that the attachment of the junctional epithelium on the microgap hinders bacterial influence on the crestal bone as well as the greater height with respect to the junctional epithelium interface, and the greater mechanical stability of the internal cone connection with respect to the external hexagon connection [4].

Recently published systematic reviews and meta-analyses evaluating the effect of prosthetic platform switching on peri-implant tissues have concluded that implants with prosthetic platform switching have less peri-implant bone loss than those with conventional platforms. Hsu et al. also appreciated that these differences increased as the thickness of keratinized mucosa augmented. Annibali et al. observed that the protective effect was greater when the horizontal mismatch of the prosthetic platform change increased. However, Annibali et al. and Strietzel et al. conveyed the interpretative difficulty of the results, given that the risk of bias and the heterogeneity of the added studies were high [7,8,9].

Therefore, despite the efforts of the scientific community, there is still no consensus on the potential benefits of the platform-switching technique on peri-implant bone maintenance, given the heterogeneity of the studies and the multiple and complex factors surrounding dental implants. Furthermore, there is no information on the influence of the platform-switching technique on other peri-implant tissues.

Hence, the objective of this systematic review and meta-analysis was to evaluate the effects over time of the platform-switching technique with respect to the conventional platform, using peri-implant bone loss and probing depth as primary variables.

## 2. Material and Methods

### 2.1. Strategic Search and Information Gathering

Two of the authors conducted a systematic search adhering to PRISMA guidelines in the electronic databases of PubMed, Embase, WebofScience, and Scopus in November 2020. Studies published between January 2010 and November 2020 were considered. An advanced search with specific terms was also performed, and randomized clinical trials (RCTs) were filtered. A screening was performed by title and by abstract, and finally after the complete reading of the articles, the studies that met the inclusion criteria were included. There was no disagreement between the two authors regarding the selection of the studies.

The participant, intervention, comparison, and outcome (PICO) format was adopted to select the studies. Thus, the PICO question was, “Compared to implants without prosthetic platform switching (C), do those with prosthetic platform switching (I) have a favorable effect on peri-implant tissues (O) in patients with implants (P)?” In this study, the results of both the prosthetic platform change (PS) group and the conventional platform (RP) group were collected and compared.

Thus, the advanced search was as follows: 

((“dental implants” [mh]) OR (“dental” [tiab] AND “implants” [tiab]) OR (“dental” [tiab] AND “implant” [tiab]) OR (“endosseous implant” [all] AND “dental” [tiab])) AND ((“dental implant abutment design” [all]) OR (“dental” [tiab] AND “implant—abutment” [tiab] AND “design” [tiab]) OR (“dental implant-abutment design” [all])) OR ((“dental” [tiab] AND “implant” [tiab] AND “platform” [tiab]) AND (“switching” [tiab] OR “platform switch” [tiab] OR “platform switching” [tiab] OR “switched platform” [tiab] OR “platform switched” [tiab] OR “platform-switched” [tiab] OR “platform mismatch” [tiab] OR “platform-mismatched” [tiab] OR “platform shift” [tiab] OR “platform shifting” [tiab] OR “platform-shifted” [tiab]))

The inclusion criteria applied to the articles were (1) English or Spanish published studies, (2) randomized clinical trials with a follow-up of at least 12 months after prosthetic loading of the implant, (3) those articles that compared the effects on the peri-implant tissue of the prosthetic platform change with respect to implants with a conventional platform, and (4) studies that reflected the necessary data on the state of the peri-implant tissues in order to perform the meta-analysis. Nine articles, all of them randomized clinical trials (RCTs), were obtained and subjected to qualitative and quantitative analysis (Figure 1).

### 2.2. Risk of Bias

The quality of the selected studies was assessed using the Cochrane randomized clinical trial assessment tool, which quantitatively indicates the risk of bias of each study according to the completion, indicated by the + sign, or noncompletion, indicated by the—sign, of seven items. Each failure in the application of these items is summed up on a scale of 0 to 7.

### 2.3. Data Collection

Data collection was performed by a single examiner and consisted of (1) type of study, (2) average age and gender distribution of patients, (3) inclusion and smoking bias, (4) type of platform (PS/PM), (5) flap design, (6) loading protocol, (7) manner of platform switching, (8) type of prosthetic restoration, (9) implant length and diameter, and (10) amount of marginal discrepancy. The clinical data obtained were (A) implant survival at 1 year, (B) peri-implant bone loss at 1 year, (C) thickness of keratinized mucosa, (D) probing depth, calculated using the Comprehensive meta-Analysis v3.0 program, (E) bleeding on probing, and (F) indexes (plaque index, gingival index, and calculus index).

### 2.4. Meta-Analysis

A statistical analysis was performed with a random effects model of the data obtained from the different studies regarding the vertical loss of peri-implant bone at 1 year and the increase in probing depth at 1 year, comparing the PS and PM groups, with the software. The existence of significance was assessed with the Z-test and *p*-value = 0.05.

While the studies presented the bone loss at 1 year, the increase in probing depth had not been calculated as they presented a first value (baseline) and a second value (at 1 year) but not their difference. Therefore, a first mean difference had to be established to calculate the increase in probing depth at 1 year for the PS and PM groups.

When the data were ready, they were entered into the Comprehensive meta-Analysis v3.0 computer program, which calculated the mean difference of the parameters chosen from all the articles, making it possible to estimate the difference in the vertical bone loss and the increase in the overall probing depth between the PS and PM groups, as well as the confidence interval of the results and statistical significance.

The heterogeneity of the studies was analyzed by the I-Squared value, which according to Higgins et al. represents low heterogeneity when I-Squared is around 25%, moderate heterogeneity when it is around 50%, and high heterogeneity when it is around 75%, and by and the Q-test, which indicates the existence of heterogeneity when the *p*-value is less than 0.1 [10].

The results were represented graphically in a forest plot, and a sensitivity analysis was carried out for the meta-analysis estimate for the studies included in it using the One Study Remove technique, which represented the difference in the means of the multiple combinations of studies, eliminating a different one each time.

## 3. Results

The present systematic review includes data from nine selected randomized clinical trials, classifying them in three different tables that analyzed different aspects of the studies: the first table analyzed the data that characterized the type of study, a sample, and the risk of bias of the study (Table 1); the second one compiled the features of the implants used in each study (Table 2); and the third one studied the different clinical parameters of the two groups of implants (Table 3).

All included articles were published between 2012 and 2019 and were randomized clinical trials (RCTs). The mean age of the participants was similar in all studies, varying between 43.29 [16] and 56.2 years of age [13]. In terms of gender distribution, the total sample included 339 females and 188 males, representing 80.3% more females than males. This difference is further accentuated in two particular studies, Telleman et al. [17] (17/0) and Telleman et al. [19] (77/15).

The inclusion criteria contemplated accepting individuals who smoked as long as they did not exceed 10 cigarettes per day, and this characteristic was present in 4 of the 9 articles.

The number of samples of implants with prosthetic platform change (PS) and without prosthetic platform change (PM) of the included studies was similar in all of them, the total implants samples being 475 and 462, respectively. The most disparate study in this regard had a sample size of 45 (PS)/25 (PM) [11]. All these data are shown in Table 1.

The characteristics of the implants used in each study is compiled in Table 3.

The application of the Cochrane Risk of Bias Assessment Tool for Randomized Clinical Trials is illustrated in Table 4.

Regarding the surgical procedure and flap design, Fernández et al. [16] did not convey this information. Thus, of the remaining eight studies, all of them performed a full-thickness flap, which was subsequently closed in two studies until prosthetic restoration [11] or until an exposed healing abutment was added 8 weeks after surgery [15]. In the remaining six studies, the flap was sutured around an exposed healing abutment that modified the gingival profile of the implant until the time of prosthetic restoration. In all of the studies, implants were placed with a minimum horizontal distance of 3 mm between implants and 1.5 mm between implant and natural tooth.

Fernández et al. [16] did not specify the implant loading protocol either. Therefore, of the remaining eight articles, six loaded the implant 3 months after surgery, as is the case with Lago et al. [12], and explained that the implants would be loaded at least 2 months after surgery and Pozzi et al. [15] loaded the implants 4 months after surgery.

With respect to whether the prosthetic restoration was screw retained or cemented, seven of the nine studies indicated that their restorations were cemented; Pozzi et al. [15] indicated theirs were cement-screw retained with TempBond^®^; and Uraz et al. [11] performed screw-retained restorations.

In reference to the dimensions of the implants used, their length and diameter were explicitly recorded in eight of the nine articles. Lago et al. [12] only referred to the brand of the implants, whose lengths varied from 6 to 18 mm and diameters from 3.3 to 4.8 mm. In the remaining eight articles, the lengths ranged from 8 to 14 mm and the diameters from 3.3 to 5 mm. Finally, the marginal discrepancy of implants with prosthetic platform change was recorded, which was taken into account in five of the articles and whose length was 0.3–0.4 mm.

The different clinical parameters of the two implant groups are collated in Table 3, both at baseline, i.e., at the time of loading, and 1 year after loading (baseline/1 year). In those cases where the data were only recorded in the 1-year sector, the variation in the parameter indicated at the year of loading with respect to the baseline (Δ1 year) will be referred to.

Survival of the implants at 1 year was quantitively high. Thus, in the only articles where it was not 100%, a survival rate of 93.1% in the case of PS and 94.5% in the case of PM was recorded [13] and 97.3% in the case of PS and 100% in the case of PM was recorded [14].

Regarding parameters reflecting peri-implant health, bleeding on probing and plaque index were recorded in seven of the nine articles, gingival index in four of the nine articles, and calculus index in three of the nine articles. The values were low and reflected a stable peri-implant health. For the studies that did not record these parameters, it was explicitly reported in the studies that they had stable peri-implant health. The peri-implant keratinized mucosal thickness was not recorded in any of the articles.

The parameters amenable to meta-analysis, vertical bone loss at year 1, and increased probing depth were recorded in nine and six of the articles, respectively. Thus, vertical bone loss at year 1 in the PS group ranged from −0.08 mm in the best-case scenario [12] to −0.68 mm in the worst-case scenario [16] and vertical bone loss at year 1 in the PM group ranged from +0.15 mm [20] to −2.23 mm [16]. The difference in the means of the two groups, PS and PM, concluded that the PM group had 0.255 mm more peri-implant bone loss than the PS group. The study by [12] was the only one where the PM group suffered less vertical bone loss than the PS group, although the difference was minimal, leading to the conclusion that there were no statistically significant differences between the PS and PM groups.

Regarding the peri-implant probing depth at year 1, the records ranged from 2.33 mm [11] to 2.2 mm [12] for the PS group and from 2.46 mm [14] to 1.7 mm [12] for the PM group. However, the difference in the means for the increase in the probing depth at year 1 for the two groups, PS and PM, revealed that in the PS group the probing depth increased by 0.19 mm more than in the PM group. In addition, the study by Guerra et al. [14] was the only one where the increase in the probing depth was greater in the PS group than in the PM group.

### Meta-Analysis

(a)Peri-implant bone maintenance:

Data from the nine included studies were combined using a random-effects model. A mean difference of −0.255 mm (95% confidence interval, between −0.437 and −0.072) was estimated with a *p*-value of 0.006 and a Z-value of −2.737, meaning that the PM group had peri-implant bone loss of 0.255 mm more on average than the PS group, the results being statistically significant. Furthermore, the heterogeneity of the studies was high, with a Q-value of 68.180, a *p*-value of 0.000, and an I-Squared value of 88.266 (Figure 2).

One Study Remove of the nine articles was performed, but there were no relevant differences (Figure 3).

(b)Increased probing depth:

Data from the six included studies were combined using a random-effects model. A mean difference of −0.082 mm (95% confidence interval, between −0.339 and 0.175) was estimated with a *p*-value of 0.532 and a Z-value of −0.625, meaning that the PS group had a probing depth of 0.082 mm more on average than the PM group, although the results are not significant. The heterogeneity of the studies was high, with a Q-value of 44.809, a *p*-value of 0.000, and an I-Squared value of 88.842 (Figure 4).

One Study Remove of the six articles was performed (Figure 4), and it was appreciated that when omitting the article by Guerra et al. [14], the difference in the means was −0.190 (95% confidence interval, between 0.216 and 0.464), with a Z-value of −3.557 and a *p*-value of 0.000, which is statistically significant. In this case, heterogeneity was greatly reduced, with a Q-value of 3.882, a *p*-value of 0.422, and an I-Squared value of 0.000 (Figure 5).

## 4. Discussion

Recently, several reviews and meta-analyses have aimed to discern the influence of prosthetic platform change on peri-implant bone maintenance. The promotion of standardized research methods to compare different types of implants by meta-analysis is a fundamental requirement to reduce the heterogeneity of the results and thus increase their relevance. 

Regarding the procedures used in the different meta-analyses, Santiago et al. proposed improvement criteria, including the use of only randomized clinical trials with standardized research methods (since some studies have recorded bone loss using conventional X-rays and others with CBCT), the use of samples in accordance with the significance of the results, the use of two groups of implants with a split-mouth design, and the establishment of systematic searches carried out in different databases and in different languages [21].

In the studies included in the present review, patients were randomly distributed. Furthermore, their selection was carefully carried out by including multiple selection criteria with the aim of excluding patients with systemic diseases, such as diabetes, osteoporosis, a history of radiotherapy, a history of bisphosphonate use, or coagulation disorders. Patients practicing harmful habits, such as smoking, excessive alcohol consumption, drug use, and poor oral hygiene, were also excluded.

The patient selection criteria and the randomization of the articles included in the present study are beneficial in terms of homogenization of the results. However, there are many other individual factors that could influence implant behavior, such as immunological, genetics, and microbiological characteristics of patients, which are not usually evaluated in this type of randomized clinical trials [1].

The surgical techniques used for implant placement were similar in all the articles included in the present study, all of them opting for the performance of a full-thickness flap. There is no indication that the type of surgery performed may have a major influence on the long-term health of dental implants. In support of this, in 2019, Singh et al. performed a review comparing the survival of implants in patients with type II diabetes placed with mucoperiosteal flap surgeries or flapless surgeries. The results were 94.2% and 92.3%, respectively; therefore, the differences were minimal, without any statistical significance [22].

With respect to the loading protocol employed, in all studies included, it was a conventional one. In this regard, in 2019, Chen et al. compared the influence on peri-implant health of immediate loading, early loading, and conventional loading protocols, concluding that there was a higher incidence of implant failure in immediate loading [23]. The most commonly used method of retention of restorations was the cemented restoration (7/9). In only one article [11] were screw-retained restorations used, and in another [15], cement-screwed-retained restorations were used. In 2017, Wittneben et al. conducted a systematic review to compare the long-term health of implants with cemented or screw-retained prostheses. The results revealed a higher number of technical and biological complications in cemented prostheses [24]. The dimensions of the implants used in the included studies ranged from 8 to 14 mm in length and from 3.3 to 5 mm in diameter. In a 2018 consensus, Jung et al. agreed that short implants (<6 mm) have a similar survival period to long implants (>6 mm) at 1 and 5 years of function and recommended a minimum diameter of 4 mm for short implants [25]. All implants used in the present meta-analysis were long implants.

Regarding bone level, in 2020, Santiago et al. carried out a review of seven meta-analyses, comparing the influence on the maintenance of the peri-implant bone level of implants with PS and PM, with the aim of comparing the results and evaluating their procedures. The quantitative analysis of the results of the different studies revealed that the peri-implant bone loss at 1 year in implants with PS was 0.29 mm less than in implants with PM. The qualitative analysis of the studies, performed with the AMSTAR scale for the evaluation of the quality of studies, resulted in a quality between 6/8 (high) and 2/8 (medium). Thus, they concluded that PS implants preserved the peri-implant bone better than PM implants [21].

Concerning the results of the meta-analysis of the present study, a greater vertical bone loss of 0.255 mm was observed in the PM group with respect to the PS group, indicating that implants with a prosthetic platform change suffer less vertical bone loss than those with a conventional platform after 1 year of loading. This result was statistically significant, although the sensitivity analysis revealed some heterogeneity. Other recent meta-analyses that have compared peri-implant bone loss in implants with a prosthetic platform change and implants with a conventional platform have obtained similar results to those of the present study. Thus, the review by Santiago et al., in 2020, which compared the results of seven different meta-analyses, revealed that peri-implant bone loss after 1 year in implants with PS was 0.29 mm less than in implants with PM [21].

The present review and meta-analysis included multiple parameters indicative of peri-implant health in the data collection. Peri-implant vertical bone loss was the most recorded parameter and the primary objective of all RCTs. The other parameters indicative of peri-implant health or disease were recorded with the aim of contextualizing the condition of the implants and their possible influence on vertical bone loss. In this sense, even in studies that did not include any additional peri-implant health parameters, reference was made to adequate implant health during the study, reflected in the high survival rates of the implants [20].

Probing depth was recorded in six of the nine included studies, being 0.082 mm greater in the PS group than in the PM group, although this result was not statistically significant and also suffered from high heterogeneity. However, when One Study Remove was performed, it was appreciated that, omitting the results of Guerra et al. (2014), the probing depth in the PS group was 0.190 mm greater than in the PM group, a statistically significant result with low heterogeneity [14]. This finding, which is apparently paradoxical, could be explained by the more apical and horizontal positioning of the supracrestal-peri-implant-attached tissue in implants with prosthetic platform change, which would result in a greater probing depth despite a smaller bone loss. No literature was found to support this result quantitatively, as peri-implant probing depth is not usually subjected to meta-analysis. Probing depths varied from baseline to year 1. These data could suggest that the probing depth stabilizes around 2–2.5 mm although when the initial measurements are less than 2 mm, substantial increases are seen. However, the final probing depths remain similar between the two groups, PS and PM.

## 5. Conclusions

(1).Implants with prosthetic platform change have less vertical bone loss during their first year of loading than implants with conventional platform.(2).The peri-implant health parameters of the two groups, PS and PM, were similar, with a healthy peri-implant condition being reported in all cases. However, statistically significant differences were found regarding the probing depths of the two groups, which was greater in the PS group than in the PM group, according to One Study Remove.(3).Further long-term studies are required to observe the maintenance of these differences over time.

## Figures and Tables

**Figure 1 jcm-11-01743-f001:**
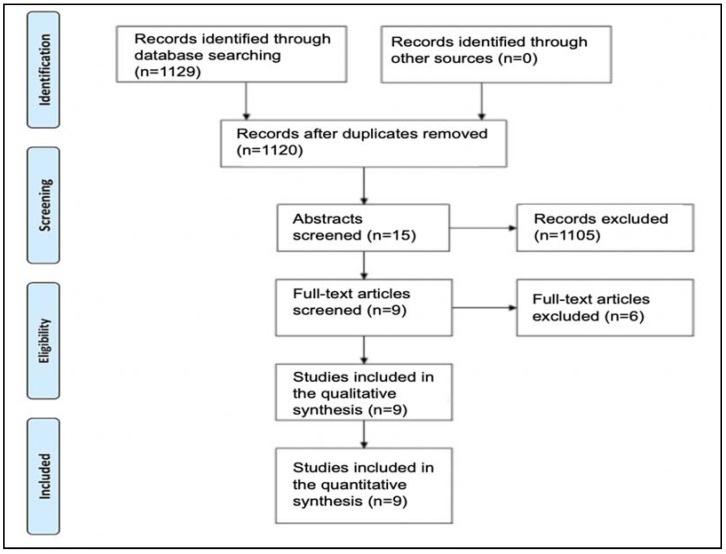
Prisma flow diagram.

**Figure 2 jcm-11-01743-f002:**
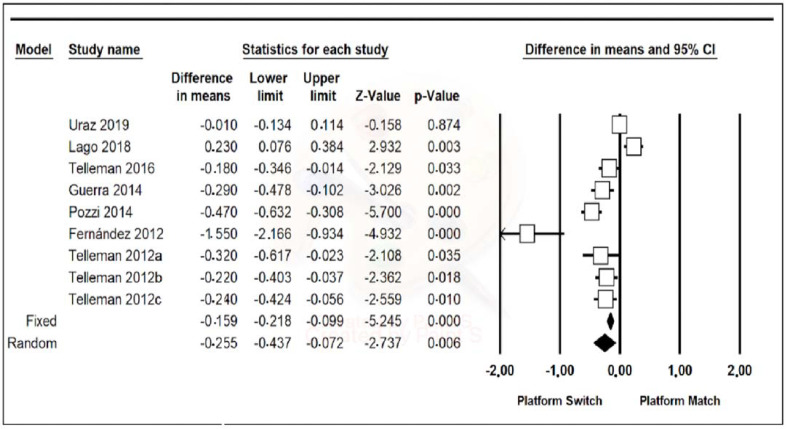
A forest plot of peri-implant bone level maintenance.

**Figure 3 jcm-11-01743-f003:**
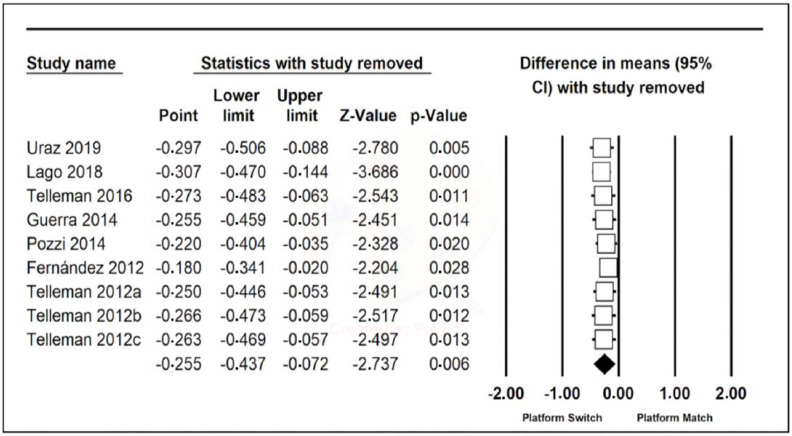
One Study Remove of peri-implant bone loss.

**Figure 4 jcm-11-01743-f004:**
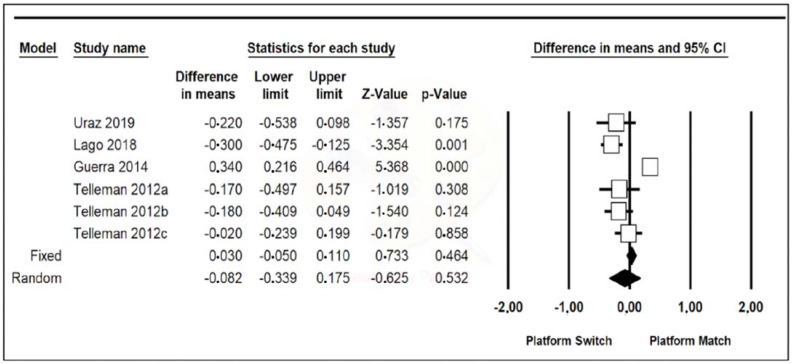
A forest plot increased probing depth.

**Figure 5 jcm-11-01743-f005:**
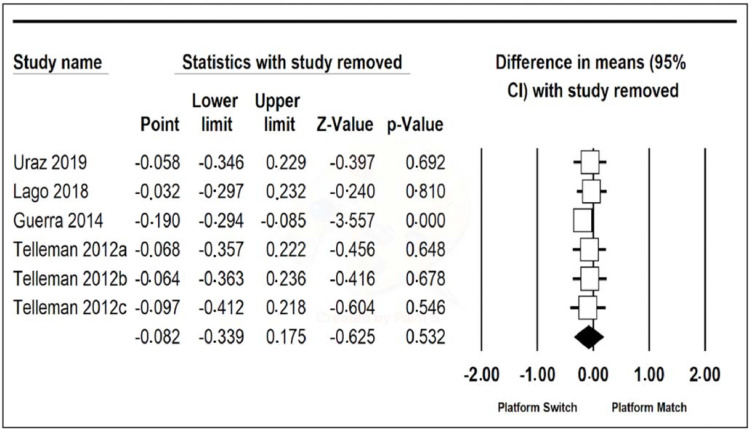
One Study Remove of increased probing depth.

**Table 1 jcm-11-01743-t001:** Sample and quality of the studies.

Reference	Type of Study	Age and Gender (F/M)	Smokers	Number of Implants (PS/PM)	Bias Risk
[11]	RCT	36/34 50 ± 7.3	Yes, up to 10 cig.	45/25	3/7
[12]	RCT	20/15 49.5	Yes, up to 10 cig.	50/50	1/7
[13]	RCT	53/27 56.2	No	56/59	3/7
[14]	RCT	31/37 51.44	Yes, up to 10 cig.	74/72	2/7
[15]	RCT	19/15 52.2	Up to 10 cig.	34/34	2/7
[16]	RCT	33/18 43.29	No	58/56	2/7
[17]	RCT	17/0 53.7	No	31/31	3/7
[18]	RCT	53/27 49.75	No	54/59	3/7
[19]	RCT	77/15 50.59	No	73/76	3/7

RCT: randomized clinical trial; cig.: cigarettes.

**Table 2 jcm-11-01743-t002:** Implant features.

Reference	Flap Design	Loading Protocol	Type of Restoration (Cemented/Screwed)	Length	Diameter	Mismatch
[11]	Closed mucoperiosteal flap	3 months	Screwed	9.5–14 mm	3.5–4.5 mm	NA
[12]	Mucoperiosteal flap with exposed healing abutment	2 months minimum	Cemented	6–18 mm (Straumann Standard Plus type model and bone-level-type model)	3.3–4.8 mm (Straumann Standard Plus type model and bone-level-type model)	NA
[13]	Mucoperiosteal flap with exposed healing abutment	3 months	Cemented	8.5 mm	4–5 mm	0.35−0.40 mm
[14]	Mucoperiosteal flap with exposed healing abutment	3 months	Cemented	9–13 mm	3.8–5 mm	0.3–0.35 mm
[15]	Mucoperiosteal flap closed and healing cap exposed after 8 weeks	4 months	Cement-screwed with TempBond^®^	8.5–13	3.9–4.1	NA
[16]	NA	NA	Cemented	8–14	3.3–4.8	NA
[17]	Mucoperiosteal flap with exposed healing abutment	3 months	Cemented	8.5 mm	4–5	0.35–0.4
[18]	Mucoperiosteal flap with exposed healing abutment	3 months	Cemented	8.5 mm	4.1–5	0.35–0.4
[19]	Mucoperiosteal flap with exposed healing abutment	3 months	Cemented	8.5	4.1–5	0.35–0.4

PS: platform switch; PM: platform match; NA: not assessed.

**Table 3 jcm-11-01743-t003:** Peri-implant clinical parameters.

Reference	Survival at Year 1	Vertical Bone Loss at Year 1 (PS/PM)	Thickness of Keratinized Mucosa	Δ Probing Depth	Δ Bleeding on Probing	Δ Plaque Index	Δ Gingival Index	Δ Calculus Index
[11]	100%	Δ1 year PS: −0.16 ± 0.25 PM: −0.17 ± 0.26	NA	Baseline PS: 2.3 ± 0.7 PM: 2.5 ± 0.85 1 year PS: 2.33 ± 0.45 PM: 2.31 ± 0.72	Baseline PS: 11.43 ± 15.5 PM: 16.66 ± 20.97 1 year PS: 15.9 ± 17.53 PM: 13.66 ± 15.12	Baseline PS: 0.4 ± 0.46 PM: 0.35 ± 0.41 1 year PS: 0.48 ± 0.42 PM: 0.46 ± 0.37	Baseline PS: 0.81 ± 0.42 PM: 0.9 ± 0.44 1 year PS: 0.91 ± 0.3 PM: 0.85 ± 0.42	NA
[12]	100%	Δ1 year PS: −0.08 ± 0.26 PM: 0.15 ± 0.49 PS: 0.14 ± 0.35 PM: 0.18 ± 0.46 (3 years)	NA	Baseline PS: 1.7 ± 0.7 PM: 2.2 ± 0.4 1 year PS: 2.2 ± 0.6 PM: 2.4 ± 0.5	Baseline PS: 0.05 ± 0.6 PM: 0.2 ± 0.3 1 year PS: 0.2 ± 0.5 PM: 0.2 ± 0.4	Baseline PS: 0.2 ± 0.5 PM: 0.4 ± 0.6 1 year PS: 0.1 ± 0.2 PM: 0.2 ± 0.4	NA	NA
[13]	PS: 93.1% PM: 94.5%	Δ1 year PS: −0.46 ± 0.43 PM: −0.64 ± 0.44 PS: −0.21 ± 0.45 PM: −0.27 ± 0.34 (5 years)	NA	NA	Δ1 year PS: 0.56 PM: 0.39	Δ1 year PS: 0.34 PM: 0.2	Δ1 year PS: 0.09 PM: 0.03	Δ1 year PS: 0.03 PM: 0.02
[14]	PS: 97.3% PM: 100%	Δ1 year PS: −0.4 ± 0.46 PM: −0.69 ± 0.68	NA	Baseline PS: 1.78 ± 0.79 PM: 1.69 ± 0.51 1 year PS: 2.21 ± 0.47 PM: 2.46 ± 0.51	Baseline PS: 0.05 ± 0.12 PM: 0.01 ± 0.06 1 year PS: 0.21 ± 0.28 PM: 0.20 ± 0.29	Baseline PS: 0.25 ± 0.46 PM: 0.06 ± 0.18 1 year PS: 0.1 ± 0.21 PM: 0.09 ± 0.18	NA	NA
[15]	100%	Δ1 year PS: −0.68 ± 0.34 PM: −1.15 ± 0.34 PS: −0.67 ± 0.39 PM: −1.24 ± 0.47 (3 years)	NA	NA	Δ1 year PS: 0 PM: 0	Δ1 year PS: 0.01 PM: 0.01	NA	NA
[16]	100%	Δ1 year PS: −0.68 ± 0.88 PM: −2.23 ± 0.22	NA	NA	NA	NA	NA	NA
[17]	100%	Δ1 year PS: −0.53 ± 0.54 PM: −0.85 ± 0.65	NA	Δ1 year PS: −0.36 ± 0.61 PM: −0.19 ± 0.72	Δ1 year PS: 0.27 PM: 0.4	Δ1 year PS: 0.27 PM: 0.51	Δ1 year PS: 0.07 PM: 0.17	Δ1 year PS: 0.0 PM: 0.0
[18]	100%	Δ1 year PS: −0.51 ± 0.51 PM: −0.73 ± 0.48	NA	Δ1 year PS: 0 ± 0.73 PM: 0.18 ± 0.5	Δ1 year PS: 0.56 PM: 0.39	Δ1 year PS: 0.34 PM: 0.21	Δ1 year PS: 0.09 PM: 0.03	Δ1 year PS: 0.02 PM: 0.0
[19]	100%	Δ1 year PS: −0.50 ± 0.53 PM: −0.74 ± 0.61	NA	Δ1 year PS: −0.22 ± 0.59 PM: −0.2 ± 0.76	NA	NA	NA	NA

Δ: increase; PS: platform switching; PM: platform match; NA: not assessed.

**Table 4 jcm-11-01743-t004:** Cochrane Risk of Bias Assessment Tool for Randomized Clinical Trials.

Reference	Selection Bias		Performance Bias	Detection Bias	Attrition Bias	Reporting Bias	Other Bias	Total
	Random Sequence Generation	Allocation Concealment	Blinding of Participants and Personnel	Blinding of Outcome Assessment	Incomplete Outcome Data	Selective Reporting	Anything Else, Ideally Prespecified	
[11]	+	+	+	+	-	-	-	3/7
[12]	+	+	+	+	+	-	+	1/7
[13]	+	+	+	+	-	-	-	3/7
[14]	+	+	+	+	-	-	+	2/7
[15]	+	+	+	+	+	-	-	2/7
[16]	+	+	+	+	+	-	-	2/7
[17]	+	+	+	+	-	-	-	3/7
[18]	+	+	+	+	-	-	-	3/7
[19]	+	+	+	+	-	-	-	3/7

(+) compliance with the strategy to avoid bias, (-) non-compliance with the strategy to avoid bias.

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
