# Peer review of "Comparative Study by Systematic Review and Meta-Analysis of the Peri-Implant Effect of Two Types of Platforms: Platform-Switching versus Conventional Platforms"

_jcm, 2022, doi:10.3390/jcm11061743_

Round 1

Reviewer 1 Report

The systematic review entitled “Comparative study by systematic review and meta- analysis of the peri-implant effect of the two types of platform: platform switching versus conventional plat- form” submitted to JCM aims to evaluate the effects of changing the prosthetic platform on peri-implant tissue after one year of prosthetic loading.

The systematic review appears interesting however this topic is extensively dealt with in the literature. I have some suggestions for improving the quality of the manuscript.

  • English language: Minor spell check is required. The grammatical part needs to be revised well.
  • Article structure - references go between [ ] and added before the end of the sentence (before the full stop). References should be listed in the bibliography following the Mendeley method.
  • Introduction: this section should be improved by making the text more fluent and avoiding listing dates etc.
  • I suggest to include PICO format in material and methods part.
  • Methods: Why did you include text in Spanish language??
  • Results: I notice many discrepancies in the articles recruited in the systematic review.

This may invalidate the results of the systematic review and meta-analysis. Several variables are not specified in the articles reviewed for the systematic review.

Although the text is very interesting and the authors have analysed the articles in detail, I believe that more attention in the systematic review is needed.

The text is too long and sometimes redundant.

In addition, other variables should be considered and listed in the text to improve the quality of the manuscript.

Author Response

Thanks a lot for your suggestions and your interest in the article. This is what have changed after the feedback of the editors:

We’ve tried to make the introduction of the article more fluent by listing dates, as they already show in the references, though we’ve added a paragraph that explains the results of similar systematic reviews and meta-analyses following another editor’s instructions (introduction, paragraph 5).

We’ve changed the form of the bibliographic references from () to [ ] and added all of them at the end of the sentence and before the full stop, as you suggested.

We’ve changed the PICO question from the introduction to the material and methods part, as you and the other editors suggested (material and methods, paragraph 2).

We are working closely with the translation team to avoid any grammar mistakes and not making errors as the inclusion of a paragraph in spanish was.

Finally, the influence of the discrepancies in the articles recruited for the systematic reviews in the outcomes of the study are discussed extensively throughout the discussion part. We included the most homogeneous articles admitting certain differences to increase the sample size. Certain variables were not added to increase the clarity of the text as almost none of the articles included contemplated them.  

Thanks a lot for your interest and valuable feedback. We hope that you like the changes made.

Reviewer 2 Report

Comment #1:

The manuscript was relevant. The text is clear and easy to understand. However, the authors should work on the English language.

Comment #2:

The authors ignored many similar systematic reviews which should include their outcomes in the introduction and compare their results to your outcome in the discussion. The following systematic reviews are an example:

  • Hsu, Yung-Ting, Guo-Hao Lin, and Hom-Lay Wang. "Effects of Platform-Switching on Peri-implant Soft and Hard Tissue Outcomes: A Systematic Review and Meta-analysis." International Journal of Oral & Maxillofacial Implants1 (2017)
  • Annibali, Susanna, et al. "Peri‐implant marginal bone level: a systematic review and meta‐analysis of studies comparing platform switching versus conventionally restored implants." Journal of clinical periodontology11 (2012): 1097-1113.
  • Strietzel, Frank Peter, Konrad Neumann, and Moritz Hertel. "Impact of platform switching on marginal peri‐implant bone‐level changes. A systematic review and meta‐analysis." Clinical oral implants research3 (2015): 342-358.

Comment #3:

The PICO was mentioned at the end of the introduction, it should be written in the part of materials and methods.

Comment #4:

The discussion is well written and clear justifying the importance of the manuscript fully. But need to compare the review outcomes with previous systematic reviews mentioned in comment 2.

Comment #5:

The conclusion is very important, so, it requires reform in both sites (at the end of the abstract and at the end of the manuscript) without mentioning values or numbers, just with a brief statement to summarized all content.

Author Response

Thanks a lot for your suggestions and your interest in the article. This is what have changed after the feedback of the editors:

We’ve included the outcomes of the systematic reviews and meta-analysis suggested in the introduction part (introduction, paragraph 5).

We’ve changed the PICO question from the introduction to the material and methods part, as you and the other editors suggested (material and methods, paragraph 2).

Finally, we’ve changed the conclusion form, both at the abstract and at the conclusion part, avoiding listing quantitative results.

Thanks a lot for your interest and valuable feedback. We hope that you like the changes made.

Reviewer 3 Report

The manuscript is entitled: "Comparative study by systematic review and meta-analysis of the peri-implant effect of the two types of plantform: platform switching versus conventional platform"

The authors tried to evaluate the effects of changing the prosthetic platform on peri-implant tissue after one year of prosthetic loading by systematic review and meta-analysis.

I have some comments and suggestions regarding this manuscript.

1. In the introduction section, the authors only mentioned "the objective of meta-analysis." They did not mention the objective of systematic review. It is better to add the systematic review and meta-analysis together.

2. Figure 1 shows that the authors follow the PRISMA guidelines, but they did not mention it in the manuscript. The authors should mention this at the beginning of the search strategy.

3. Was this systematic review and meta-analysis registered in PROSPERO? If the authors register, they should add the PROSPERO number in the materials and methods section.

4. How many authors performed the search strategy? The authors should mention. What about the disagreement between the authors when performing the search strategy?

5. The authors have clearly mentioned the inclusion criteria in the materials and methods section. My recommendation is to add the exclusion criteria not in the discussion section, just after the inclusion criteria.

6. "The bias mean risk of the articles analyzed was 2.44/7." I could not get the point. How does it arrive? The risk of bias assessment should be included in a different headline and not under the search strategy. The value should be added in the results section.

7. The results and discussion sections are described clearly.

The manuscript has a significant impact on the dental society. 

Author Response

Thanks a lot for your suggestions and your interest in the article. This is what have changed after the feedback of the editors:

1- We corrected the mistake that you noted regarding the objective of the article at the end of the introduction part (introduction, paragraph 7).

2- We now mention that we follow the PRISMA guidelines at the search strategy part (material and methods, paragraph 1)

3- This systematic search and meta-analysis was not registered in PROSPERO.

4- Now the article mentions how many authors performed the systematic search and if there was any disagreement (material and methods, paragraph 1).

5-There are no exclusion criteria in the present study, only the inclusion criteria shown in the material and methods part. During the discussion part, the article talks about the multiple inclusion and exclusion criteria of the randomized clinical trials analyzed, and how they could be improved, and how they could affect the outcome of the study.

6- We’ve added the bias risk table at the end of the results of the systematic review as you suggested. In the material and methods part, We’ve added a paragraph that explains the functioning of the Cochrane Risk of Bias Assessment Tool for Randomized Clinical Trials (material and methods, paragraph 6). Finally, we’ve avoided adding “The bias mean risk of the articles analyzed was 2.44/7”.

Thanks a lot for your interest and valuable feedback. We hope that you like the changes made.
